
# 1 Importance of Biogenic Volatile Organic Compounds to Peroxyacyl

# 2 Nitrates (PANs) Production in the Southeastern U.S. during SOAS 2013

Shino Toma[1], Steve Bertman[1], Christopher Groff[2], Fulizi Xiong[2], Paul B. Shepson[2], Paul Romer[3],
Kaitlin Duffey[3], Paul Wooldridge[3], Ronald Cohen[3], Karsten Baumann[4], Eric Edgerton[4], Abigail R.
Koss[5,7*], Joost de Gouw[5], Allen Goldstein[6], Weiwei Hu[7,8], and Jose L. Jimenez[7,8]
[1]{Department of Chemistry, Western Michigan University, Kalamazoo, MI, USA}
[2]{Departments of Chemistry, and Earth, Atmospheric, and Planetary Sciences, Purdue University, West
Lafayette, IN, USA}
[3]{Department of Chemistry, University of California, Berkeley, CA, USA}
[4]{Atmospheric Research & Analysis, Inc., Cary, NC, USA}
[5]{NOAA ESRL Chemical Sciences Division, Boulder, CO, USA}
[6]{Department of Environmental Science, Policy and Management, University of California, Berkeley,
CA, USA}
[7]{Cooperative Institute for Research in Environmental Sciences, University of Colorado, Boulder,
Colorado, USA}
[8]{Department of Chemistry and Biochemistry, University of Colorado, Boulder, Colorado, USA}
*now at Department of Civil and Environmental Engineering, Massachusetts Institute of Technology,
Cambridge, MA, USA
Correspondence to: S. Bertman (steven.bertman@wmich.edu)
KEY WORDS: isoprene, PAN, MPAN, SOAS, BVOC

## 24 Abstract

Gas-phase atmospheric concentrations of PAN, PPN, and MPAN were measured at the ground using GC-
ECD during the SOAS 2013 campaign (1 June to 15 July 2013) in Centerville, Alabama in order to study
biosphere-atmosphere interactions. Average levels of PAN, PPN and MPAN were 169, 5, and 9 pptv
respectively, and the sum accounts for an average of 15% of $NO_y$ during the daytime (10 am to 4 pm local
time). Higher concentrations were seen on average in air that came to the site from the urban $NO_x$ sources
to the north. PAN levels were the lowest observed in ground measurements over the past two decades in
the Southeastern U.S. Analysis of PAN/$NO_x$ indicates PAN production in this region was sensitive to $NO_x$.
A multiple regression analysis indicates that biogenic VOCs account for 66% of PAN formation in this
region of the Southeastern U.S. Comparison of this value with a 0-D model simulation of peroxyacetyl
radical production indicates that at least 50% of PAN formation is due to isoprene oxidation. MPAN has a
statistical correlation with isoprene hydroxynitrates (IN) with an average $MPAN_{ppb}$/$IN_{ppb}$ ratio of 0.3.
Organic aerosol mass increases with gas-phase MPAN and IN concentrations, but the mass of organic





nitrates in particles is largely insensitive to MPAN. Isoprene and PAN play a significant role in the
atmospheric chemistry in the Southeastern United States.
**1.   Introduction**
The Southern Oxidant and Aerosol Study (SOAS), part of the Southeast Atmosphere Study (SAS) in 2013,
focused on biosphere-atmosphere interactions and the subsequent influences of biogenic volatile organic
compounds (BVOCs) on atmospheric oxidant chemistry and aerosol formation. Since natural emissions of
organic compounds in the Southeast are high and there are a variety of anthropogenic pollution sources, the
location is ideal to study biogenic-anthropogenic interactions (Carlton et al., 2018).
Peroxyacyl nitrates (PANs, $RC(O)OONO_2$), products of the photooxidation of VOCs in the presence of
nitrogen oxides ($NO_x$), play an important role in gas and particle tropospheric chemistry. PANs account for
a significant amount of $NO_y$ in rural and forested areas in the Eastern United States (Trainer et al., 1993),
affect the lifetime of $NO_x$, enhance $O_3$ formation (Carter et al., 1981), contribute to human health issues,
and are known to be phytoxic (Finlayson-Pitts and Pitts, 2000; Kleindienst et al., 1990).
Peroxyacetyl nitrate (PAN) is the simplest and most abundant of the PANs and peroxypropionyl nitrate
(PPN) and peroxymethacryloyl nitrate (MPAN) also are observed in the field (e.g. Nouaime et al., 1998;
Pippin et al., 2001; Roberts, 2002). PAN is formed from both anthropogenic and biogenic hydrocarbon
precursors. PPN, on the other hand, is formed primarily from anthropogenic hydrocarbons (AHCs) (e.g.
propanal, propane, 1-butene) while MPAN is derived from methacrolein (MACR), an oxidation product of
the mostly biogenic hydrocarbon (BHC), isoprene (Biesenthal and Shepson, 1997; Carter and Atkinson,

20  1996).

Recent laboratory experiments have suggested that OH reaction with the double bond of MPAN could be
involved in the formation of secondary organic aerosol (SOA) (Chan et al., 2010; Kjaergaard et al., 2012;
Lin et al., 2013; Nguyen et al., 2015; Surratt et al., 2010; Worton et al., 2013). This pathway is currently
treated in a few models that include isoprene (e.g. Pye et al., 2013; Pye et al., 2015; Jenkin et al., 2015;
Wennberg et al., 2018), although because isoprene is the biogenic non-methane hydrocarbon with the
greatest global emission rate (Guenther et al., 1995), the contribution of isoprene photooxidation to aerosol
radiative forcing may be underestimated. We currently do not know the partition coefficient of MPAN with
the particle phase, and little about condensed phase chemistry of MPAN.
We measured PANs concentration during the SOAS 2013 campaign to characterize the systematic
behavior and levels of individual PAN species at an urban-impacted forest and to assess the current state of
the attribution of PANs formation to biogenic and anthropogenic precursors quantitatively using several



statistical methods. Finally, we compared MPAN with another nitrogen compounds in the gas phase, total
isoprene hydroxynitrates (IN) and with organic nitrates or total organic aerosol (OA) in the particle phase
to investigate relationships that might explain their influence on SOA formation.
**2. Experimental**
Ground-based measurements were conducted from 1 June to 15 July 2013 at the Southeastern Aerosol
Research and Characterization (SEARCH) Centreville (CTR) site, which was located in a forested area in
the Talladega National Forest near Brent, Alabama, (lat: +32°54′11.81″, long: -87°14′59.79). The major
anthropogenic influence at this site comes from the cities of Tuscaloosa and Birmingham, which are located
50 km northwest and 80 km northeast respectively.
Measurements of PANs using similar methods to those described below were made in Dickson, TN from
15 June to 14 July, 1999 as part of the Southern Oxidants Study (SOS) (Cowling et al., 1998) and are
referred to in the text. The site is located near Montgomery Bell State Park about 60 km west-southwest
(upwind) of downtown Nashville (Chen, 2001). While in a different part of the southeast, we believe that
the distance from major urban areas makes this site a good comparison.
PANs were quantified using a custom gas chromatograph (GC) equipped with a Shimadzu GC-Mini-2
$^{63}$Ni electron capture detector (ECD) maintained at 55 °C (described by (Nouaime et al., 1998)). A polar
column (RESTEK, Rtx-200, 15 m x 0.53 mm ID x 1 μm) was kept at 15 C° to minimize thermal
decomposition of PAN compounds. Helium was used as carrier (8 cm$^3$ min$^{-1}$) with $N_2$ make-up gas (3 cm$^3$
min$^{-1}$). Ambient air was drawn through a ¼″ OD PFA Teflon tube from 8.2 m height above the ground at 1
SLPM and a sub-sample of this air was drawn through a 1 cm$^3$ sample loop at 50 sccm. The residence time
was approximately 9 sec. The sample loop contents were injected into the column at 20 min intervals via a
6-port Teflon valve (Hamilton). The baseline and sensitivity of the GC-ECD were checked every day during
the campaign using standard addition of gas streams from liquid standards added to ambient air scrubbed
through a charcoal trap at the beginning of the sampling line. In this way, the impact of the inlet line was
accounted for in the calibration. Separate calibrations were performed with synthetic PAN, PPN, and
MPAN samples in dodecane or tridecane maintained at ice water temperature in diffusion cells. The level
of PAN in each synthetic compound was determined with a chemiluminescence $NO_x$ analyzer (Themo
Environmental Instruments, Inc., Model 42S) equipped with a Mo converter held at 325°C. The converter
efficiency was tested by $O_3$ titration of NO to $NO_2$. Calibration of the $NO_x$ analyzer was done against a
NIST-traceable cylinder of 2ppmv NO in $N_2$ (SCOTT-MARRIN, INC). Based on sensitivity and
background measurements, the detection limits (S/N=2) for PAN, PPN and MPAN were estimated as 2.5





pptv, 3.6 pptv and 3.9 pptv, respectively. Uncertainty determined by error propagation, most of which came
from the chemiluminescence $NO_x$ analyzer, was estimated to be 20% RSD.
Measurements of other trace gases, such as $NO_y$, $NO_x$, and $O_3$, wind direction, temperature, and boundary
layer height were made by Atmospheric Research & Analysis, Inc. (ARA) as described by Hidy et al. (2014).
Total isoprene hydroxynitrate (IN) concentrations were determined by Purdue University using a chemical
ionization mass spectrometer (CIMS) with operating conditions described by Xiong et al. (2015). MACR
was measured by NOAA ESRL Chemical Sciences Division and University of California, Berkeley
(Goldstein group) using a GC-MS. Particle-phase organic nitrates (pONs) were measured by University of
California, Berkeley (Cohen group) using thermal dissociation laser-induced fluorescence (TD-LIF),
described by Rollins et al. (2010), and by University of Colorado with a high resolution time of flight
aerosol mass spectrometer (HR-ToF-AMS) described by DeCarlo et al. (2006) and Hu et al. (2015). A
comparison in Lee et al. (2016) found that the pONs-TD-LIF was generally higher by factor ~5 than pONs-
HR-ToF-AMS. Both sets of data provide a reasonable range of pONs concentration. Total OA mass was
measured using HR-ToF-AMS.  Data below detection limit were excluded from statistical analysis.
**Comparison of PANs measurements among WMU, ARA, and UC Berkeley**
During the SOAS 2013 campaign, two other research groups measured the sum of total PANs without
identification of each species. ARA measured total PANs using thermal dissociation into $NO_2$ at 160 °C on
top of ambient $NO_2$ located within 30 m of the WMU instrument and at the same height. The Berkeley
group measured total PANs using thermal dissociation from the tower approximately 100 m north of the
WMU instrument and approximately 25 m above the ground. Total PANs from all three groups showed
statistically significant ($p < 0.01$) positive linear correlations with each other based on results from
Spearman's rank correlation test (a nonparametric test was used due to non-normal distributions). The
correlation coefficient, $r_s$ of each pair ($PANs_{WMU}$ vs. $PANs_{ARA}$, $PANs_{WMU}$ vs. $PANs_{UC}$, and $PANs_{ARA}$ vs.
$PANs_{UC}$) was 0.754, 0.926, and 0.714 respectively. However, a Friedman test resulted in statistically
different medians of PANs from three groups. The relationships with $PANs_{WMU}$ are plotted in Figure S1.
Overall, the measurement of $PANs_{UC}$ was 50% greater than $PANs_{WMU}$, while the measurement of $PANs_{ARA}$
was 30% less than $PANs_{WMU}$. The strong statistical correlation of all datasets allows the investigation of
PANs behavior despite the systematic differences.
**3.  Results**
**3.1      General behavior of PANs in 2013**





Figure 1 shows a time series of PAN, PPN, and MPAN throughout the campaign. Data that were below
detection limit (BDL) are plotted at half of the reported detection limit for that compound. This was done
to distinguish the BDL points from missing data due to tests, calibrations, and the periodic existence of a
noise interference that often appeared during this campaign and could not be eliminated, and to not lose the
low concentration information content. Relatively high levels of PAN were observed as periodic spikes
during the campaign, but overall PAN levels were lower than most other measurements in the southeast
made over the last 20 years. A local biomass burning event was observed on June 4[th] (Washenfelder et al.,
2015), which resulted in an unusually high level of PAN of around 1600 pptv and an extreme deviation
from the median. Hence, the data on June 4[th] was removed from statistical analyses.
General descriptive statistics for all daytime data are summarized in Table 1. Daytime was defined as 10
am to 4 pm local time (CDT). PAN was consistently the most abundant peroxyacyl nitrate compound and
the mean concentration during daytime was 34 and 19 times larger than that for PPN and MPAN,
respectively. In Table 1, "PANs" describes the sum of individual PAN, PPN, and MPAN values. The
average of the ratio of PANs/NO$_y$ during daytime was 0.15. Peroxyacryloyl nitrate (APAN) was also
observed occasionally during the campaign. APAN has been proposed to arise from 1,3-butadiene, either
from anthropogenic sources or biomass burning, and from direct emission of acrolein (Roberts et al., 2001;
Tanimoto and Akimoto, 2001). Our data did not show a strong relationship to biomass burning events, as
identified by Washenfelder et al. (2015), although an instrument interference problem limited the amount
of reportable APAN data, so no clear conclusion can be drawn.
Although surface air most frequently came from the south during the SOAS 2013 campaign, air from the
north contained levels of PANs that were twice as large as from south. The averages of PAN, PPN, and
MPAN with air from the north were 182, 5.3, and 8.4 pptv respectively, while averages of air from south
showed 94.6, 2.8, and 3.6 pptv. Polar plots of PAN, PPN and MPAN as a function of surface wind direction
are shown in Figure S3 with wind frequency. This elevated northern distribution is also seen with NO$_x$ and
O$_3$ reflecting the influence of anthropogenic pollution sources from Tuscaloosa, Birmingham, and Atlanta.
Plots of diurnal mean and median values separated by surface wind direction (Figure 2) indicate a
noticeable pattern in PAN, PPN, and MPAN from the north and a much weaker pattern in southerly air.
Levels of all three PANs were highest (also with greatest variance) during the daytime on average. It was
difficult to see the mean and median diurnal cycle for PPN because of very low concentrations observed
over the campaign. The PAN diurnal pattern was generally similar to those reported for Nashville in 1995
and 1999 (Nouaime et al., 1998; Roberts, 2002). A calculation using ambient temperature and [NO]/[NO$_2$]
shows that the effective PAN thermal decomposition lifetime changes little over the course of the afternoon,





which suggests that PAN levels fluctuate during early afternoon mostly due to dilution by boundary layer
growth (boundary layer height increased by a factor of 2-3 from 9am to noon-3pm on average based on a
LIDAR measurements).
**3.2     Historical PANs measurements in the Southeastern US over last 23 years**
PAN compounds have been measured at various rural and urban locations within the Southeastern U.S.
over the last 23 years. Observations from six sites, Elberton (GA) 1990; ROSE (AL) 1990 and 1992; New
Hendersonville (TN) 1994; Youth Inc. (TN) 1995; Dickson (TN) 1999; Cornelia Fort Airpark (TN) 1999
(Nouaime et al., 1998; Roberts, 2002; Roberts et al., 1998) are compared here with SOAS 2013 data (a map
of the locations is shown in Figure S2).
Binned PAN concentrations during the daytime (10 am – 4 pm) are plotted as a function of the concentration
of $NO_x$ (grouped into deciles) in Figure 3a. Urban areas have higher PAN concentration with higher $NO_x$
than rural areas. The only site specially revisited was ROSE, where PAN levels in 1990 were more than
twice as high as in 1992. Hence, the PAN concentrations can vary depending on place and year. A curve fit
to the data in Figure 3a shows an asymmetric peak in the concentration that appears to peak at around 3.5
ppb $NO_x$. PAN concentration increases approximately linearly with $NO_x$ up to 2 ppb and beyond the peak
it decreases slowly with further increases in $NO_x$. Similar behavior was observed in the relationship between
$O_3$ and $NO_x$ concentration in Figure 3b and the peak was at around 1.5 ppb $NO_x$.
The relationship of $O_3$ production with $NO_x$ and VOC concentrations is typically discussed in terms of
"$NO_x$-limited" and "VOC-limited" regimes, (Finlayson-Pitts and Pitts, 2000; Milford et al., 1994;
Chemeides et al., 1992), although there has been less discussion of the sensitivity of PAN production to
these reactants. This curve is reminiscent of the modeled $O_3$ production rate as a function of $NO_x$ and $HO_x$
in Thornton et al. (2002) from OH oxidation of VOC based on measurements from Cornelia Fort Airpark
in 1999. At low NO concentration, an increase of $O_3$ production rate with NO occurred, since OH is
generated via $HO_2$+NO and the primary chain termination are $HO_x$+$HO_x$ reactions. On the other hand, $O_3$
production rate slows at high NO concentrations. In this $NO_x$-saturated (VOC-limited) regime, OH is
consumed, because $HO_x$+$NO_x$ reactions and $RO_2 + NO \rightarrow RONO_2$ (Romer et al., 2016) become faster than
$HO_x$+$HO_x$ reactions. The peak is related to the crossover point between $NO_x$-limited and $NO_x$-saturated. A
high $HO_x$ production rate (radical-limited conditions) enhances the $O_3$ production rate with NO for low NO
and the crossover point shifts to higher NO.
As a significant covariance between PAN and $O_3$ has been reported (Bottenheim et al., 1994), the behavior
of the relationship of PAN vs. $NO_x$ could result from similar sensitivity of PAN production as $O_3$ production.



A linear increase of PAN concentrations with $NO_x$ at low $NO_x$ in Figure 3a could result from the chemistry
in the $NO_x$-limited regime and most PAN concentrations at rural sites were dependent on $NO_x$
concentrations. The slow decrease of PAN concentration at higher $NO_x$ levels such as those seen at more
urban sites results from faster radical termination rates, and thus VOC oxidation rates slow. This
empirically-derived distinction is likely related to differences in reaction rates with peroxy radicals that
could be investigated computationally. PAN, $O_3$ and $NO_x$ levels in the Southeast were all lowest at SOAS
2013. As $NO_x$ levels continue to decrease in the US with more stringent emission standards and conversion
to non-fossil energy sources, PAN production rate might become more widely sensitive to $NO_x$.
**3.3       Anthropogenic vs Biogenic contribution to PAN production**
**3.3.1   Description of MLR and its Statistical Meaning**
A multiple linear regression (MLR) has been used to quantify PAN sources (Roberts, 2002; Roberts et al.,
1998; Williams et al., 1997). Since the thermal decomposition rates of PANs are similar (Roberts and
Bertman, 1992), and MPAN and PPN are formed from BHC and AHC respectively and PAN is formed
from both, [PAN] can be approximately represented as a weighted linear combination of [MPAN] and
[PPN]. The combination of BHC and AHC chemistry is indicated by MPAN and PPN. The linear model is
applied as in equation 1.
$[PAN] = A + B_1[MPAN] + B_2[PPN]$                                                                  (1)
Here, $A$ is the intercept and $B_1$ and $B_2$ are partial regression coefficients, estimated using a computer
software program based on field observations. The MLR statistical analysis includes estimation of $A$, $B_1$
and $B_2$, overall $F$-test and a $t$-test, and diagnostic procedures (e.g. Mendenhall et al., 2008). The $F$-test is
used to investigate the statistical significance of the model in Equation (1) using an analysis of variance
(ANOVA) table. The strength of the model is evaluated using the coefficient of determination $R^2$ between
predicted and measured [PAN] (also provided through this statistical analysis). The individual $t$-test, which
is based on the Student's t statistic, is used to investigate the statistical significance of the individual $B_1$ and
$B_2$. In a MLR statistical analysis, the magnitude of the standardized partial regression coefficients, $\beta_i$, which
is calculated as a product of partial regression coefficient and the ratio between the standard deviation of
the respective independent variable (MPAN or PPN) and the standard deviation of the dependent variable
(PAN), is frequently used to compare the relative contribution of independent variables. The results of MLR
statistical analysis are summarized in Tables S1 and S2.
Tatsuoka (1971) showed that $R^2$ from the MLR is equal to the sum of the product of the $\beta_i$ and the zero-
order (simple bivariate) correlation, $r_i$, which are obtained as results of MLR (see Table S2). That is, $R^2 =$





$\Sigma \; \beta_i r_i$. Therefore, we used the fraction of $R^2$ based on the strength of relationship in each [MPAN] and
[PPN] to [PAN] to describe the relative importance of BHC and AHC. Each partial $R^2$ is obtained as shown
in equations 2 and 3.
$R^2_{BHC} = \beta_1 r_{MPANvs.PAN}$                                                    (2)
$R^2_{AHC} = \beta_2 r_{PPNvs.PAN}$                                                    (3)
This approach allows us to directly treat the $R^2$ in the MLR to assess the relative importance of BHC and
AHC, including the strength of correlation with PAN.
Results from SOAS were compared with similar PAN data collected from Dickson, TN in 1999, another
rural southeastern site, which show that the MLR model and regression coefficients for both MPAN and
PPN at both sites were statistically significant (see Tables S1 and S2). During SOAS 2013, 60% of the
variance in the measurements was explained by the MLR model. At the Dickson site in 1999, 77% of the
variance was explained by the MLR model. The $R^2$ of MLR in the SOAS 2013 data was lower than that in
Dickson 1999, which might result from the lower absolute PANs levels during SOAS 2013. In particular,
SOAS MPAN and PPN data included a large number of below detection limit measurements, while Dickson
1999 data did not. The medians of PAN, MPAN, and PPN in Dickson 1999 (483.5, 25.4, and 24.7 pptv
respectively) were three times higher than the medians for SOAS 2013. Also, in Dickson 1999, $NO_x$ levels
were seven times higher. In Figure 4, the relative importance of BHC and AHC was standardized to compare
SOAS 2013 and Dickson 1999. Standardized relative percentiles were calculated as $R^2_{BHC}/R^2 \times 100$ for BHC
and $R^2_{AHC}/R^2 \times 100$ for AHC. Biogenic influence accounted for 66% of PAN during SOAS 2013 and was
two times larger than the anthropogenic influence. This is the opposite of results from Dickson where the
biogenic influence accounted for only 25% of PAN. Although both sampling locations were located in rural
areas in similar environments, the results using MLR indicate that the HC precursors were different.
**3.3.2    Comparison of contribution of isoprene oxidation with computational modeling**
We used two computational approaches to assess the contribution of isoprene oxidation to PAN formation
by 1) orthogonal distance regression (ODR) between field measurements of MPAN and PAN and 2)
simulation of the production of peroxyacetyl (PA) radicals, precursors of PAN, using an ambient 0-D
photochemical model.
In ODR, the fraction of PAN production from isoprene oxidation can be expressed as
$(d[PAN]_{isoprene}/dt)/(d[PAN]_{all}/dt)$. Assuming that MPAN is solely derived from isoprene oxidation, the
relative yield of $(d[MPAN]/dt)/(d[PAN]_{isoprene}/dt)$ was obtained from an isoprene oxidation chamber



experiment as 0.15±0.03 RSD. The reaction was initiated with 1.37 ppm isoprene, 268 ppb NO, and 206
ppb $NO_2$ under 5% RH in a 5.5 $m^3$ Teflon cylindrical bag. The OH radical was produced by photolysis of
HONO. The fraction of PAN production from isoprene oxidation to total PAN formation in the field was
rearranged as the following equation.
$$\frac{d[PAN]_{isoprene}/dt}{d[PAN]_{all}/dt} = 6.7 \times \frac{d[MPAN]/dt}{d[PAN]_{all}/dt}$$       (4)
The $(d[MPAN]/dt)/(d[PAN]_{all}/dt)$ was obtained from measurements in SOAS 2013 as the slope of the linear
regression line of [PAN] to [MPAN].
In the 0-D photochemical model simulation, the relative contribution to peroxyacetyl (PA) radicals from
VOCs present at the field site is based on the Master Chemical Mechanism (MCM) v3.3. The ambient
model included not only isoprene and its oxidation products (including $CH_3C(O)CHO$) but also acetone,
acetaldehyde, and some mono-terpenes as precursors. The detail of the parameters for MCM set are
described in Groff (2015).
These two methods were compared with the relative importance of BHC, $\beta_1 r_{MPANvs.PAN}$, from the MLR
model in this work. Four days (Jun 3[rd] 12:30 – 18:00, Jun 14[th] 11:30 – 18:00, Jun 26[th] 11:00 – 18:00, and
Jul 12[th] 13:00 – 18:00) of data from SOAS 2013 were selected to run the 0-D model because the production
ratio using ODR can only be used when PAN, MPAN and $NO_x$ concentrations were appropriately high.
The time ranges were chosen so that the boundary layer height would be stable and any dilution effect
would be minimal. Results of the comparison are plotted in Figure 5. Although the relative importance of
BHC in the MLR model was less than 40% on June 3[rd], it was statistically dominant on the other three days
accounting for more than 68%. (Note: PPN on June 3[rd] did not have a significant level to predict PAN in
MLR analysis, p=0.600.) By comparison, the estimated contribution of isoprene oxidation using ODR on
June 3[rd] had the steepest slope, however, the range of the 95% confidence interval (C.I.) on this day was
large. On the other three days, the relative contributions of isoprene oxidation using ODR were estimated
at 23 – 49%, lower than the results derived from the other two methods. This might be due to the differences
between the chamber experiment and ambient conditions. Specifically, the ratio of secondary to primary
oxidation products varies between chamber and ambient conditions, with likely relatively greater primary
products under chamber conditions. Since PAN is formed via the oxidation of secondary products of
isoprene, the $[MPAN]/[PAN]_{isoprene}$ ratio would be higher in the chamber experiment than in ambient
environment. This ratio is expected to derive results that are biased low, when used to estimate the ambient
isoprene-derived PAN concentration. In addition, photolysis rates also are significantly different between
the chamber and the field conditions. The results of the 0-D model suggest that isoprene oxidation
significantly contributed to PAN formation with a mean range of 55–73% over all selected days. The



relative contribution of isoprene oxidation determined by PA radicals was typically 7–25% lower than by
MLR model analysis on three days, except June 3rd. Hence, both methods, the MLR and the 0-D model,
indicate that isoprene oxidation was the main source when high levels of PAN were observed during SOAS

4   2013.

**4.  Discussion**
**4.1     Comparison among MACR, IN, and MPAN**
MACR is a first generation product of isoprene photooxidation mechanisms, and MPAN is derived from
MACR oxidation (Bertman and Roberts, 1991; Kjaergaard et al., 2012). With enough $NO_x$, the OH adduct
of isoprene that is the precursor to MACR in these mechanisms is also a precursor of gas-phase isomers of
isoprene hydroxynitrates (IN) (Shepson, 2007; Grossenbacher et al., 2001, 2004; Barker et al., 2003; Paulot
et al., 2009; Lockwood et al., 2010). Xiong et al. (2015) reported IN at SOAS, which affords the opportunity
to study this aspect of $NO_x$ sensitivity of isoprene oxidation. In this work, the daytime (10 am – 4 pm)
relationships among MPAN, MACR and IN at SOAS was investigated using the Pearson's correlation
statistical test. Missing data was treated as pairwise deletion (not listwise deletion). The correlation
coefficient between the first generation products of isoprene, MACR and IN was 0.528 (p < 0.001) and
indicated a statistically significant positive correlation. According to the known chemical pathways, a strong
relationship is expected between MACR and MPAN, while a weak relationship is expected between IN and
MPAN given that IN is a primary product, while MPAN is secondary. The results show, however, that
daytime data over the whole campaign did not show significant correlation between MACR and MPAN ($r$
$= 0.148$, $p = 0.104$). In contrast, IN has a statistically significant positive correlation with MPAN ($r = 0.499$,
$p < 0.001$). The average ratio of $MPAN_{ppb}/IN_{ppb}$ was 0.3 ($\pm$ 0.04), based on the slope of $[MPAN_{ppb}]$ vs.
$[IN_{ppb}]$ ($p< 0.001$, with a slope significantly different from 0). Likely this relationship is a result of the $NO_x$
dependence of both organic nitrate products. Because MACR can be produced in the absence of $NO_x$,
MPAN is more dependent on $NO_x$ than on MACR at this site, and isoprene nitrates constitute a larger
fraction of gas-phase organic nitrates from BVOC than MPAN does. This is consistent with the Romer et
al. (2016) work showing that IN production is the dominant sink for both radicals and $NO_x$ in the daytime
at this site.

**4.2     Gas-phase MPAN vs. organic aerosol mass – is there evidence that MPAN leads to more**
**organic mass and does the nitrogen from MPAN stay in particle?**




During SOAS 2013, Lee et al. (2016) estimated that the particle-phase organic nitrates (pONs) accounted
for 3% of total organic aerosol (OA) mass, on average, during the day (12 pm – 4pm) and BVOC precursors
strongly impacted the diel trends of pONs. Laboratory experiments suggest that MPAN can play a key role
in SOA formation under high $NO_x$ conditions. C4-hydroxynitrate-PAN or hydroxymethyl-methyl-$\alpha$-
lactone (HMML) (Kjaergaard et al., 2012; Nguyen et al., 2015; Wennberg et al., 2018) have been proposed
as key precursors for uptake into the particle-phase from MPAN oxidation. Nguyen and co-workers (2015)
estimated the SOA yield as approximately ~60 % by mole from MPAN + OH reaction in the absence of
$NO_x$. IN is also expected to contribute to SOA formation (Jacobs et al., 2014). If organic nitrates are
involved in SOA formation, gas-phase MPAN and IN should be related to particle mass, although the
nitrogen could be unretained in the particle. Figure 6 shows the relationship of gas-phase MPAN and IN
with daytime particle measurements to investigate if nitrogen from these organic nitrates was retained in
particles and if they are correlated with total OA without organic nitrates.
As MPAN and IN concentrations increase, Figure 6 shows that OA mass increases, while pONs mass
increases very little (measured by both HR-ToF-AMS and TD-LIF), although the slopes of MPAN and IN
vs. pONs are statistically different from zero. The relative magnitude of the response of OA and pONs to
increases in MPAN and IN suggests that they contribute to OA growth more strongly than to pONs growth.
Although this may suggest that if MPAN oxidation by OH is involved in particle growth, the nitrogen from
MPAN is not represented in aerosol organic nitrate. Oxidation of MPAN modeled from MACR + OH results
at FIXCIT (Nguyen et al., 2014) using measured total peroxynitrates and kinetics of the isoprene mechanism
in MCM v3.3.1 (Jenkin et al., 2015) showed a positive relationship between MPAN oxidation and pONs
formation, although with a yield <3% (P. Romer, personal communication). Results of direct reaction of
MPAN + OH suggest that it is unlikely that pONs formation is mainly derived from MPAN+OH reaction,
even in the presence of $NO_x$ (Nguyen and Wennberg, personal communication). This small contribution of
isoprene oxidation compounds to pONs formation is consistent with reported modelling of pONs formation
(Xu et al., 2015; Ayres et al., 2015; Pye et al., 2015). Ayres et al. suggest that pONs formation at SOAS
was dominated by nighttime reactions of $NO_3$ radicals with BVOCs rather than daytime reactions, and more
from monoterpene oxidation than isoprene oxidation.
Rather, MPAN is likely a precursor to low vapor pressure products that undergo aerosol uptake (perhaps as
a HMML precursor). The correlation between INs and MPAN with OA likely reflects that much of the OA
derives from BVOC oxidation, and the conditions that lead to large rates of BVOC emission and oxidation
(high T and radiation) simultaneously produce OA, along with INs and MPAN. That OA does not correlate
well with condensed phase organic nitrate reflects the fast hydrolysis of organic nitrates in the aerosol phase



at low aerosol pH (Rindelaub et al., 2016; Jacobs et al., 2014; Guo et al., 2015). While organic nitrates such
as the INs may partition to the aerosol phase, they are quickly converted to inorganic nitrate ion and other
products by hydrolysis.

## 5. Conclusions

The level of PAN compounds measured at the ground site during SOAS 2013, is lower than data measured
in the Southeastern U.S. over the past two decades. We show here that PAN concentrations are highly $NO_x$
sensitive. Russell et al. (2012) show that $NO_x$ in the eastern US has been decreasing rapidly, due to effective
emission control, and thus this has effectively also decreased PAN production. As this process continues,
PAN may continue to be a smaller fraction of $NO_y$, as peroxy radicals such as $CH_3C(O)OO$ react with $HO_2$
and $RO_2$ rather than with $NO_x$. PAN appears to be most sensitive up to $[NO_x]$ approximately 3.5 ppb, above
which PAN concentrations switch to a $NO_x$-saturated (or VOC-limited) regime. So, PAN production during
SOAS was highly sensitive to $NO_x$ concentration and this was particularly observed in the biogenically
derived MPAN formation. Overall, MPAN did not show a statistically significant correlation with MACR,
but did show a statistically positive correlation with IN. These results indicate that both organic nitrate
products were $NO_x$ dependent (MPAN being more sensitive to $NO_x$ than to MACR precursor) and IN
production might be the dominant sink for both radicals and $NO_x$ in the daytime at this site.
We estimate that biogenic precursors, particularly isoprene, account for about 66% of PANs, twice as much
as anthropogenic influence during the overall campaign and that gas-phase MPAN shows significant
contribution to OA growth, but less contribution to pONs during the daytime. This may suggest that the
nitrogen of MPAN is removed during oxidation to other low vapor pressure products, consistent with the
HMML mechanism first suggested by Kjaergaard et al (2012).

## 6. Acknowledgments

This research was supported by EPA STAR grant 83540901-0-RD. We thank the people of SOAS 2013
campaign. We thank Drs. James Roberts, David Parrish, Eric Williams, and Martin Buhr for providing
NOAA data of PANs and other trace gases from Elberton, ROSE, Henderson, and Cornelia Fort ground
sites. We thank Tran Nguyen and Paul Wennberg for sharing data from FIXCIT. WWH and JLJ were
supported by NSF AGS-1243354 and EPA STAR 83587701-0. This publication was developed under
Assistance Agreements awarded by the U.S. Environmental Protection Agency. It has not been formally
reviewed by EPA. The views expressed in this document are solely those of the authors and do not





necessarily reflect those of the Agency. EPA does not endorse any products or commercial services
mentioned in this publication. NSF grant AGS-1352972 supported Cohen group work at UC Berkeley.

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



Table 1. General descriptive statistics of PANs and other trace gases during SOAS 2013 campaign.
(Data on June 4[th] is not included.)

|  |  | All day | Daytime (10am–4pm) |
| --- | --- | --- | --- |
| PAN | Number of data | 2813 | 719 |
|  | Mean ± STD (ppt) | 126 ± 110 | 169 ± 129 |
|  | Median | 99 | 148 |
| PPN | Number of data | 2402 | 534 |
|  | Mean ± STD (ppt) | 4 ± 5 | 5 ± 7 |
|  | Median | 2 | 2 |
| MPAN | Number of data | 2346 | 512 |
|  | Mean ± STD (ppt) | 5 ± 7 | 9 ± 10 |
|  | Median | 2 | 2 |
| $O_3$ | Mean ± STD (ppb) | 26 ± 13 | 34 ± 11 |
|  | Median | 25 | 33 |
| $NO_x$ | Mean ± STD (ppb) | 0.6 ± 0.6 | 0.3 ± 0.2 |
|  | Median | 0.4 | 0.3 |
| PANs/$NO_y$ | Mean ± STD | 0.11 ± 0.07 | 0.15 ± 0.08 |
|  | Median | 0.10 | 0.14 |

STD means standard deviation.
PPN and MPAN include data of below detection limit, 1.8 and 1.9 pptv respectively.
PANs = PAN + PPN + MPAN





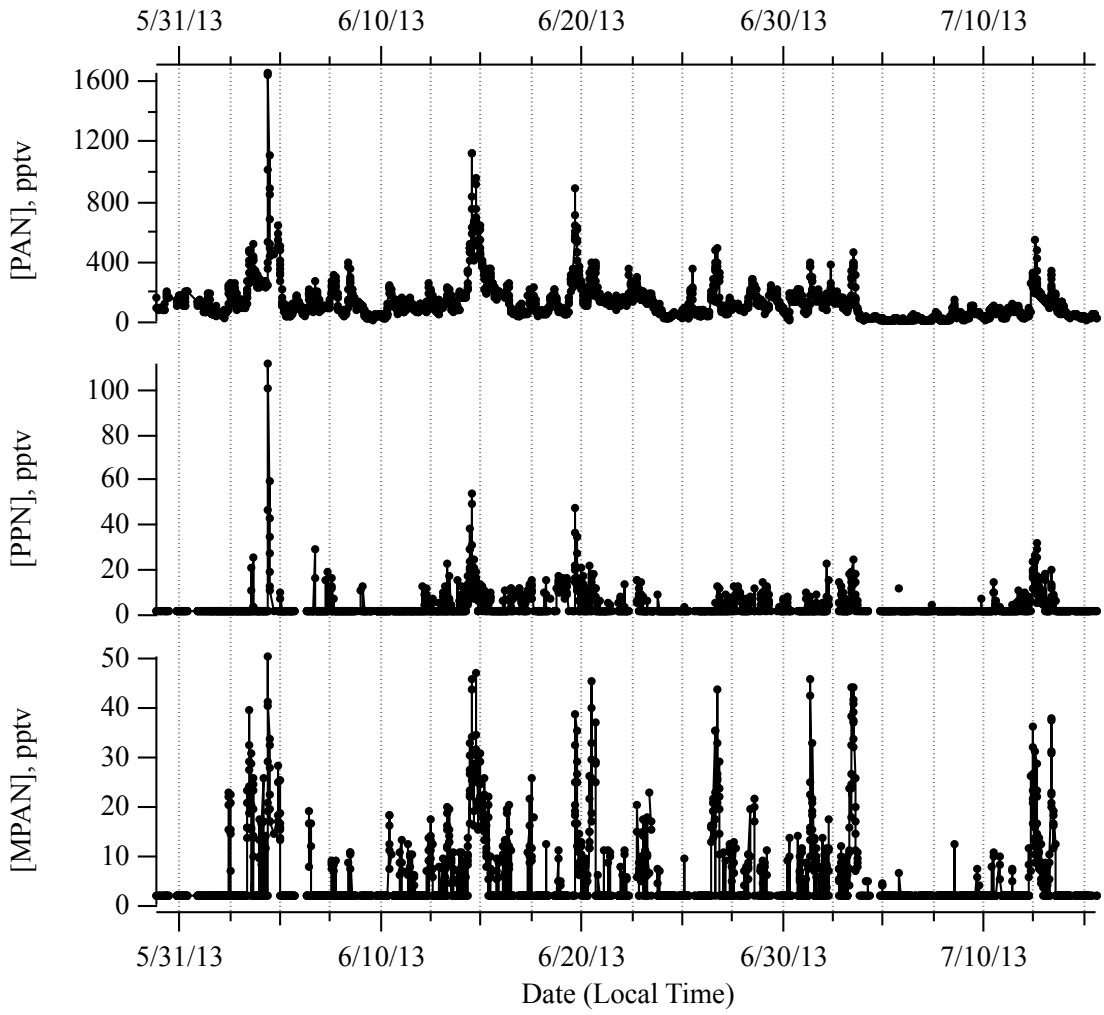

2     Figure 1. Time series of detected PAN, PPN, and MPAN during SOAS 2013 campaign using GC-ECD.



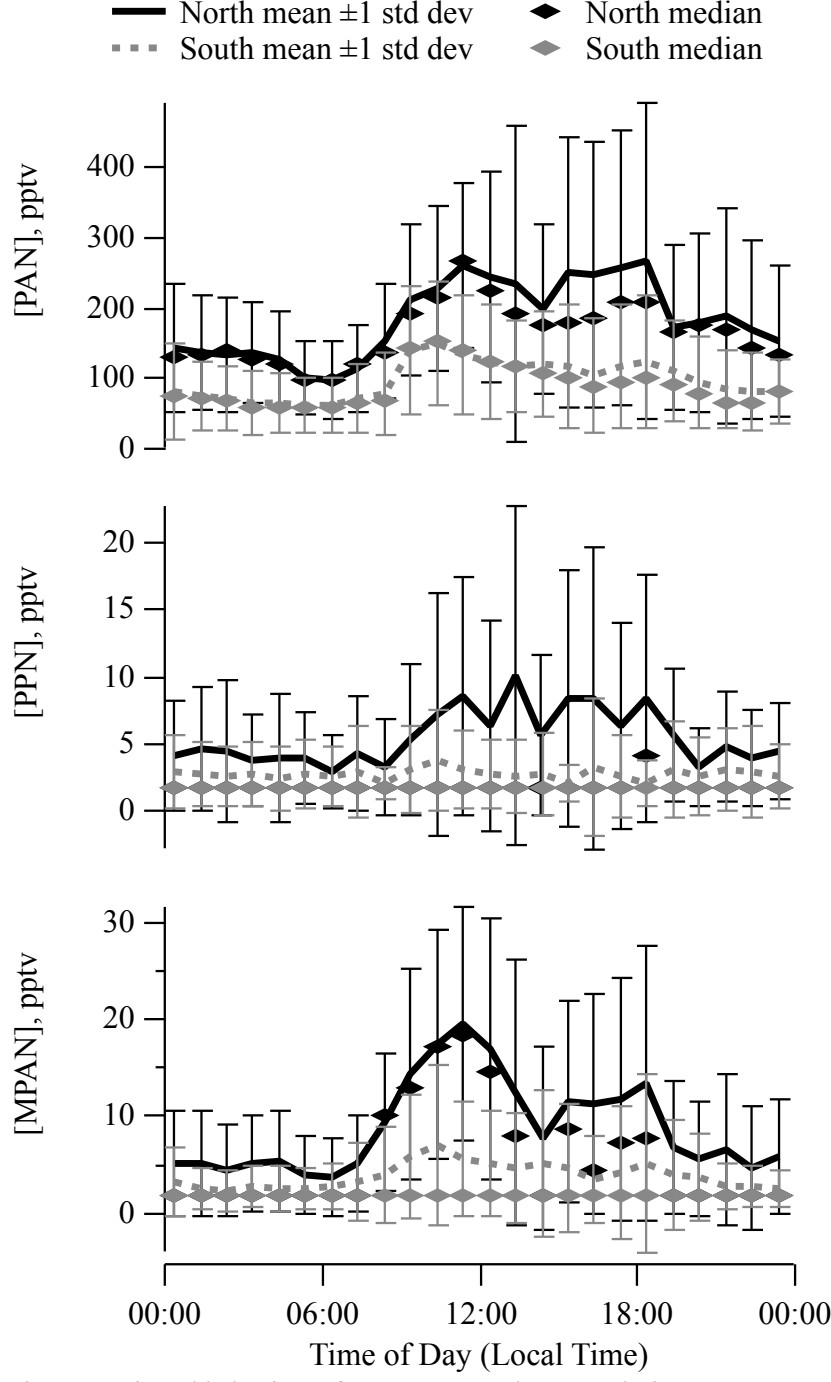

2    Figure 2. Diurnal behaviors of PAN, PPN, and MPAN during SOAS 2013 with wind from South and
3    North. Data that were below detection limit (BDL) are included at half of the detection limit. The
4    medians of PPN and MPAN from the south were consistently BDL.





Figure 3. Surface (a) PAN and (b) ozone concentrations for each ground site in the Southeastern U.S.
over the last 23 years for 10 am – 4 pm as a function of the Concentration of $NO_x$ in deciles. The solid
line indicates a fit line for all measurements.



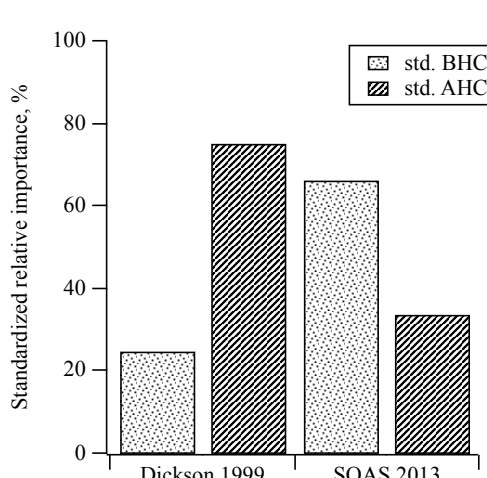

Figure 4. Comparison of standardized relative contribution to PAN formation from biogenic and
anthropogenic hydrocarbons during the daytime in Dickson, TN in 1999 and SOAS, in Centreville, AL
in 2013. The std. BHC and std. AHC mean that standardized relative importance of biogenic
hydrocarbon and anthropogenic hydrocarbon respectively.



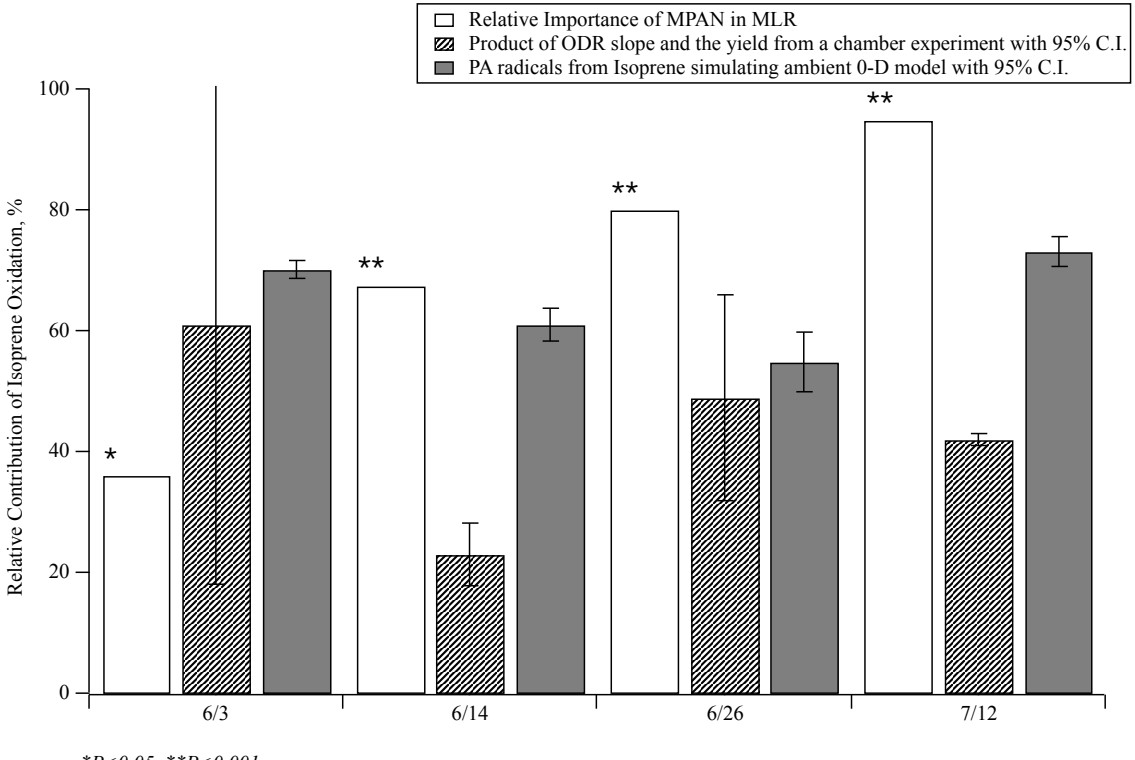

*P<0.05, **P<0.001

Figure 5. Estimates of the relative contribution of isoprene oxidation to PANs formation during 4
specific days of SOAS 2013 using three different approaches: multiple regression analysis, ODR with
chamber data, and simulation of PA radicals using a 0-D model. The *P* indicates the significant level of
t-test. The C.I. means confidence interval.





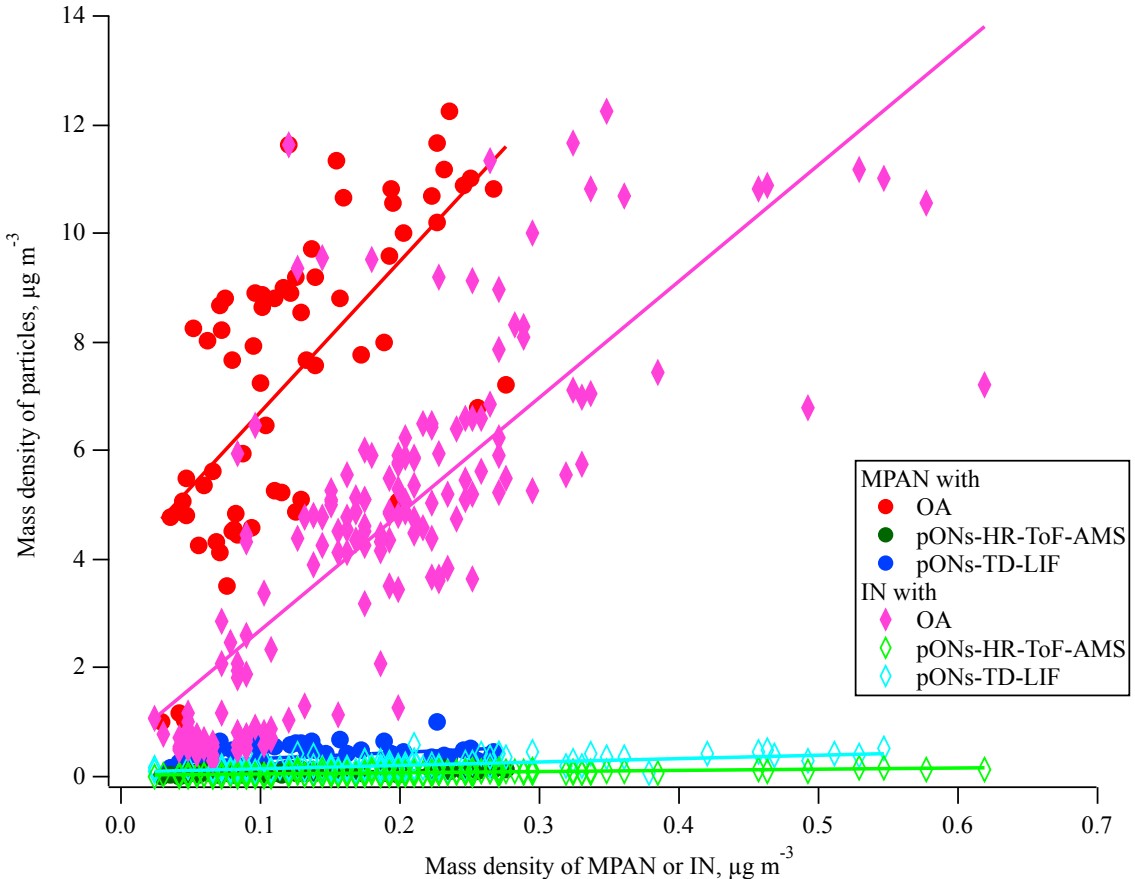

Figure 6. The relationship of mass between MPAN or IN in the gas phase and in organic aerosol for 10
am – 4 pm during June 29[th] – July 15[th] in SOAS 2013 (the time period when data on pONs-TD-LIF was
available). OA is organic aerosol (without organic nitrate) and pONs is particle-phase organic nitrates.
MPAN has a linear slope ($R^2$) of 27.8 (0.455) vs. OA, 0.4 (0.437) vs. pONs-HR-ToF-AMS, and 1.1 (0.120)
vs. pONs-TD-LIF. IN has a linear slope ($R^2$) of 21.4 (0.606) vs. OA, 0.2 (0.603) vs. pONs-HR-ToF-AMS,
and 0.6 (0.341) vs. pONs-TD-LIF.