# Peer review of "Importance of Biogenic Volatile Organic Compounds to Peroxyacyl"

_Atmospheric Chemistry and Physics, 2018_

## Referee Comment (RC1) · Anonymous Referee #1 · 11 Aug 2018

Toma et al. present measurements of PAN, PPN, and MPAN utilizing the GC-ECD technique during SOAS. Measurements of these 3 compounds by the same technique have been made in various parts of the world over the past decades. As such, analytical approaches (i.e. to attribute PAN to anthropogenic and biogenic sources using PPN and MPAN, respectiely) are also well established. Toma et al. do well to follow in those footsteps and present their work in a succint, clearly written manuscript. The authors, however, missed an opportunity to clearly deliver a more impactful (in my opinion) message, that is, on the changes in atmopsheric composition (in this case, PANs) in the backdrop of decreasing NOx emissions over the last few decades. The sections discussing the correlation between isoprene nitrate and MACR, and the correlation be-

tween MPAN and OA, sound painfully stretched to suggest the potential of something important but lack convincing evidence. The focus, in my opionion, should be on presenting the high quality of the measurements and how atmospheric composition (PANs in this instance) have changed over time, and why. The manuscript has a lot of potential and deserves consideration for publication, but after significant improvements.

Results shown on figure 3 (PAN behavior with NOx) are intruiging. But how would you discount the possibility that the plateauing/leveling-off of PAN with NOx above 3.5 ppb is not due to lack of VOC or that measurements were made so close to NOx emission sources (since the NOx levels are so high) that the VOCs did not have time to react to form PAN? Possible to utilize your 0-D model accounting for as much of observations (VOCs, NO, NO2, etc.) to detrermine the chemical scenarios under which this PAN vs NOx behavior can be reproduced? By presenting PAN versus NOx for all the campaigns (figure 3), authors are implying that NOx level is all you need to know to get PAN levels. This needs to be justified.

Results shown on figure 4 is fascinating. I think a more thorough discussion of this PAN source attribution comparison between Dickson and SOAS is merited. (Side note, making this into 2 pie charts using the same red and blue color scheme for SOAS and Dickson as in figure 3 would be nice, not critical though). For instance, WHY is anthropogenic a much bigger contributor to PAN during Dickson than SOAS? Can you look into biogenic and anthropogenic VOC emissions inventories for the two regions during the appropriate time periods to determine how they have changed? The NOx level during Dickson (figure 3) would suggest it is well below the 3.5 ppb threshold. As such, Dickson is still clearly in the "NOx limited" regime. So why would PAN attribution (anthro vs biogenic) be so different between SOAS and Dickson? There are obviously many variables that affect PAN ambient levels (boundary layer height, transport time from main regional NOx source, etc.). But at minimum presenting the approximate VOC (anthro and biogenic) precursor levels that affect PAN production and temperature that affect PAN lifetimes for the two campaigns would be helpful.

I would like to see (in the SI or main) the diurnal plot of the PAN/NOy ratio (like figure 2 of Roberts 2002) for the Dickson and SOAS campaigns. That ratio can tell you amongst other things how efficiently PANs are being produced. How have the ratios changed over time? Why?

I would like to see (in the SI) the MPAN vs PAN and PPN vs PAN scatter plots. Are the slopes comparable to obserations from other studies? Can these slopes be used as characteristic signatures of anthropogenic and biogenic influences? Roberts used a range of MPAN vs PAN depending on time of day. Did the authors have to do that as well or were the slopes constant throughout SOAS?

I would like to see (possibly appended to figure 1) a diurnal plot of the MLR calculated PAN next to the observed PAN. The MLR model explains 60% of the variance. On average, when is the agreement good and bad. I am skeptical how robust the anthro vs biogenic attribution is for SOAS given how PAN, PPN and MPAN appear all well correlated in time (figure 1).

Section on attribution of BHC derived PAN to isoprene. For the ODR approach, your only input was from chamber oxidation of isoprene. Were you able to test MPAN production rates from MBO oxidation? Monoterpenes oxidation? I am concerned that your answer came out to be isoprene only because your input to the ODR was isoprene.

I question the relevance/validity of discussion section 4.1. Authors used MPAN vs PAN to establish that 50-70% of SOAS PAN is biogenic. You determined that biogenic PAN is mostly from isoprene. So why is it surprising that MPAN and isoprene hydroxy nitrates are well correlated?

The statement that "...MPAN shows significant contribution to OA growth...", based entirely on correlation, is unconvincing. The contribution to OA from MPAN can be estimated knowing the volatility (or C*) and its abundance in the gas phase or its SOA yield along with its lifetime. More data (calculations, model runs, etc.), not citations to other papers that merely suggest the possibility, is needed.

Need citations for sentence on lines 16-18, page 10.

---

## Referee Comment (RC2) · Anonymous Referee #2 · 17 Aug 2018

Toma et al. present measurement of PAN, PPN and MPAN obtained during a ground based field cam-paign in 2013 in Southeastern U.S. The measurements were done using GC-ECD which is a well-established technique. The way the compounds were measured and the instruments deployed and cal-ibrated are well presented. The general behavior of the measured compounds is briefly described and compared with older measurements in this region.

The results and discussion sections are focused on two major points:

- The analysis of historical and recent PAN and NOx datasets which tend to show that the recent measurements correspond to a regime where PAN production is limited by

the availability of NOx.

- The comparison of different methods to identify PAN organic precursors and the use of those meth-ods on two datasets obtained in similar environment but 20 years apart. The more recent measure-ments had PAN production dominated by biogenic volatile organic compounds while the old meas-urements had PAN production dominated by anthropogenic volatile organic compounds.

The writing style is satisfying, the paper well referenced. The introduction part should be improved as it is in its current state lacunar. While the results are very interesting, the conclusions drawn could be made more solid if the authors were exploring further the different datasets and more exhaustive in their way to discuss the results.

I recommend publication after the following comments are addressed:

General comments:

Section 1: The first paragraph of the introduction should be placed in the experimental section as it is related to the measurements site and context. It would be useful to give typical mixing ratios (or rang-es) of PANs expected in urban/rural/forested environments with information on seasonal variations and global trends over the last decades. The different sinks of PANs should be described. The introduction should better underline the importance of understanding PANs chemistry (NOx removal/ transport, aerosol aging).

Section 2: The experimental section should contain a brief description of the meteorological conditions at the site during the campaign (temperature, humidity ...). Were those conditions expected at this lo-cation or were there different from classically encountered conditions? A citation could be placed if other papers describe the campaign in more details. I don't understand the point of discussing in such details the sum of PANs measurements by thermal dissociation as those data are used marginally in the paper. The authors should precise what they mean by NOy and NOy measurements.

This last comment apply to the whole manuscript.

Section 3.1: The time series of PANs (figure 1) should be accompanied by temperature, NOx and O3 as correlations are strongly expected and this would help to see anthropogenic influence. The profiles of PANs could be discussed further. To what is due the morning peak, is it advection? Mixing with residual layer? It would be useful to add a profile for global radiation to figure 2. Since the sum of PANs have been measured, it would be interesting to know what fractions of total PANs represent PAN, PPN and MPAN. This would justify the big paragraph of total PANs measurements comparison in the experi-mental section.

Section 3.2: Figure 3 is very interesting and show well that the 2013 measurements might correspond to a shift to a NOx limited regime, however, the authors should discuss further the possibility that the observations on the 2013 can be explained by a lower PHOx (due to lower photolysis rates or OH pre-cursors) as mentioned very/too briefly and explained in Thornton et al. 2002. The authors should also discuss the possibility that observations correspond to older air masses or higher NOx/PANs loss rates compared to the other measurements which would explain the lower PAN and NOx data.

Section 3.3: The authors should describe briefly the hypothesis that are necessary to apply the MLR (and not only cite references) and the validity limits. Especially, this method imply that all PANs are only lost by thermal decomposition, however, the authors state in Section 4 that the reaction MPAN + OH represent a non-negligible sink of MPAN. Doesn't it invalidate/limit the MLR analysis? LaFranchi et al. 2009, using a steady state method for PAN sources attribution, show that the results (relative parts of BHC and AHC) are strongly dependent on temperature, because this factor affects the emissions of isoprene (Worton et al. 2013). This could maybe explain the differences between the 2013 and 1999 results (if temperature was much higher in 2013). The authors should comment on this. Why not apply all 3 methods to all the historical measurements? This would allow to make the conclusions more ro-bust if for all rural

measurements in the 90s, the BHC role as PAN precursors was higher than for urban measurements, and if all rural measurements in the 90s were all more oriented toward AHC than the 2013 results. This would maybe allow to conclude on the decrease of AHC role in PANs production.

Section 4: As stated by the authors, the relationship between IN and MPAN should be depending on the NO/NO2 ratio. This fact could be well visualized by a plot of IN vs MPAN color coded with the NO/NO2 ratio. The authors cite Worton et al. 2013 but do not mention that in this reference, the au-thors state that MPAN uptake on aerosols results in the formation in organo-sulfates in the aerosol pahse, which is a likely explanation for the weak dependence of pONs on MPAN.

Section 3 and 4: Isoprene has a central role in all results and discussions but nothing is said about its mixing ratios which were measured in 2013 and 1999 together with the PANs.

Specific comments

page 1 line 31 : sensitive could be replace by "controled by NOx"

page 2 line 1 & 2 : the last sentence is not really necessary

page 2 line 9 : also give official IUPAC name

page 2 line 11 : The role and importance of PANs could be placed in a global context instead of being eeduced to the Eastern U.S.

page 2 line 13 : phytotoxic

page 2 line 14 : how abundant in term of fraction of PANs ?

page 2 line 25 & 26 & 27 : this sentence is confusing, maybe removing the because would help. aerosol radiative forcing could be replaced by secondary organic aerosol formation.

page 3 line 6 : "was" should be replaced by "is" ; which type of vegetation does "forested" correspond to ?

page 3 line 10 : what about BHC sources at the Dickson site

page 3 line 15 : what about air mass origins during the measurements ?

page 4 line 3 : what is NOy, is it total NOy by catalytic conversion ?

page 4 line 15 : what does WMU stands for ?

page 4 line 18 : Where are those WMU measurement described ? With which type of instrument ?

page 4 line 27 & 28 : the correlation allows for the investigation of PANs behavior but those values are never used in this paper, why compare the 3 instruments then ?

page 5 line 7 : add "(see Sect.3.2)" after "the last 20 years"

page 5 line 9 : what about the 2 others peaks near 1 ppbv

page 5 line 11 : what is the ratio between PAN and sum of PNs ?

page 5 line 14 : define NOy

page 5 line 20 : replace "surface air" by "air masses" or "sampled air" ; they seems to be a net differ-ence between air masses from the south and air masses from the north. Would the MLR for PAN pre-cursors identification reveal a difference as well between south and north.

page 5 line 24 & 25 : it would be nice to be able to see NOx and O3 somewhere. And what about dif-ferences in terms of VOCs mixing ratios between North and South, some BHC and AHC were measured during SOAS 2013. They could be described somewhere in the paper.

page 5 line 29 : to solve that a different scale could be used on figure 2 for the North and the South data

page 5 line 31 : maybe an equation would be helpful for the decomposition lifetime calculation.

page 6 line 11 : replace "higher PAN concentration with higher NOx" with "higher PAN and NOx con-centrations"

page 6 line 12 : what does "revisited" means ?

page 6 line 13 : "hence, the PAN concentrations can vary depending on place and year" could be re-placed by "Overall, the PAN concentrations were strongly variable between sites and years" ; which type of curve fit ?

page 6 line 17 : "and the peak was at around" should be replaced with "with a maximum around"

page 6 line 30 : yes covariance has been observed but that is because they are both produced by pho-to-oxidation of VOCs in the presence of NOx, this is the reason why PAN vs NOx looks similar to O3 vs NOx : their production pathways are the same.

page 7 line 2 & 3 : the sentence "and most PAN concentration at rural sites were dependent on NOx concentrations" is confusing and does not bring any information. It seems to say that only in rural ar-eas are PAN concentrations correlated with NOx concentrations which is not the case.

page 7 line 7 : references ?

page 7 line 11 : "sources" should be replaced with "precursors"

page 7 line 18 : it should be mentioned somewhere that the A factor correspond to background PAN

page 8 line 16 : replace "Also, in Dickson 1999 . . . higher" with : "while NOx levels were seven times higher".

page 8 line 19 : "(mostly isoprene)" could be added behind "Biogenic influence"

page 9 line 8 : describe the 0D model method in more details. What are the hypothesis ?

page 10 line 25,26,27 : the fact that IN is high during the day does not mean that its production is high during the day, it could be produced by NO3 oxidation of isoprene and have a long lifetime enough to be observed during the day, which is why talking about daytime in line 26 is not very accurate. Moreo-ver, saying that IN is the dominant sink is as well not accurate. IN is a sink of NOx if IN removal leads to a net loss of NOx, but what happens if IN releases NOx due to oxidation or due to uptake to the aerosol phase and subsequent release of NOx. How does the general context of those measurements compare to Romer et al. ?

page 11 line 4 : Worton et al. 2013 suggest that uptake of organics following MPAN + OH reaction oc-curs through the formation and subsequent uptake of methacrylic acid epoxide (MAE).

page 12 line 5 : remove "data"

page 12 line 6 : you showed than PAN production is limited by NOx availability,

page 12 line 8 : what is seen is that lower NOx emissions seem to result in lower ambient PAN concen-trations.

page 12 line 11, 12, 13 : the first part of the sentence just repeats line 6 and 7, the second part of the sentence is confusing, where is MPAN production rate as a function of NOx discussed ?

page 12 line 16 : same remark as for line 25, 26, 27 page 10, it is not clear that IN is a net sink, since the removal pathway that is discussed in this paper, aerosol uptake, does not seem to trap NOx in the aerosol phase.

page 12 line 17 : is 66% an average of the three methods ? The sentence "twice as much as anthropo-genic influence during the overall campaign" is redundant. If bio-genic influence is 66%, then the rest is obviously anthropogenic influence and logically

33% which is ... twice less.

Additional references :

LaFranchi, B. W., Wolfe, G. M., Thornton, J. A., Harrold, S. A., Browne, E. C., Min, K. E., Wooldridge, P. J., Gilman, J. B., Kuster, W. C., Goldan, P. D., de Gouw, J. A., McKay, M., Goldstein, A. H., Ren, X., Mao, J., and Cohen, R. C.: Closing the peroxy acetyl nitrate budget: observations of acyl peroxy nitrates (PAN, PPN, and MPAN) during BEARPEX 2007, Atmos. Chem. Phys., 9, 7623-7641, https://doi.org/10.5194/acp-9-7623-2009, 2009.

---

## Author Comment (AC1) · 17 Oct 2018

*We are so grateful to both reviewers for their time and insightful comments. Both reviewers were very thorough in their reading of the submitted manuscript. They made thoughtful comments and some excellent suggestions to improve the document and pushed us to think about aspects in different ways. The paper is significantly better as a result of their very careful review. Most of their suggestions have been incorporated in a revised manuscript as described below. Additions, revisions, and other changes made in the document are highlighted in the Word file for easy reference.*

**Suggestions from and responses to Anonymous Referee #1**

Results shown on figure 3 (PAN behavior with NOx) are intruiging. But how would you discount the possibility that the plateauing/leveling-off of PAN with NOx above 3.5 ppb is not due to lack of VOC or that measurements were made so close to NOx emission sources (since the NOx levels are so high) that the VOCs did not have time to react to form PAN? Possible to utilize your 0-D model accounting for as much of observations (VOCs, NO, NO2, etc.) to detrermine the chemical scenarios under which this PAN vs NOx behavior can be reproduced? By presenting PAN versus NOx for all the campaigns (figure 3), authors are implying that NOx level is all you need to know to get PAN levels. This needs to be justified.

*The reviewer is right; there are many possible inputs such as loss rates or photochemical aging or HOx production rates that could be responsible (likely a combination) for the behavior. In all these examples NOx chemistry and lifetime are key. The Figure is not intended to be seen as a mechanistic explanation but more an observation of a trend seen in a range of sites in the region. We have tried to revise the section of the text to tone down the impression that NOx level can be predictive of PAN levels on its own.*

Results shown on figure 4 is fascinating. I think a more thorough discussion of this PAN source attribution comparison between Dickson and SOAS is merited. (Side note, making this into 2 pie charts using the same red and blue color scheme for SOAS and Dickson as in figure 3 would be nice, not critical though). For instance, WHY is anthropogenic a much bigger contributor to PAN during Dickson than SOAS? Can you look into biogenic and anthropogenic VOC emissions inventories for the two regions during the appropriate time periods to determine how they have changed? The NOx level during Dickson (figure 3) would suggest it is well below the 3.5 ppb threshold. As such, Dickson is still clearly in the "NOx limited" regime. So why would PAN attribution (anthro vs biogenic) be so different between SOAS and Dickson? There are obviously many variables that affect PAN ambient levels (boundary layer height, transport time from main regional NOx source, etc.). But at minimum presenting the approximate VOC (anthro and biogenic) precursor levels that affect PAN production and temperature that affect PAN lifetimes for the two campaigns would be helpful.

*NO$_x$ levels continue to decrease in the country (Blanchard et al., 2012; Russell et al., 2012; USEPA). Emission inventories for anthropogenic VOC emissions have steadily decreased in the southeast over the last 15 years (USEPA) This is consistent with anthropogenic VOC measurements made at various SEARCH sites (including Centreville) in the SE over the same time period. In contrast, the more limited BVOC measurements made at SEARCH sites show consistent BVOC levels over the same time (Hagerman et al., 1997; Hidy et al., 2014). Table 1 now shows that mean levels of O$_3$ and NOx were substantially higher at Dickson than at Centreville, while isoprene levels are a factor of 2 higher at Centreville during the daytime. We have added a diurnal plot of isoprene concentrations from both sites in Figure S4b.*

I would like to see (in the SI or main) the diurnal plot of the PAN/NOy ratio (like figure 2 of Roberts 2002) for the Dickson and SOAS campaigns. That ratio can tell you amongst other things how efficiently PANs are being produced. How have the ratios changed over time? Why?

*We plotted the hourly mean diurnal profiles of PANs/NO$_y$ from the Dickson 1999 and SOAS 2013 campaigns in Figure S4a. This figure shows similar behavior at both sites with similar mean values (0.136 for Dickson and 0.155 for SOAS 2013, see Table 1 in the manuscript) that peak during the daytime. Roberts et al. (2002) reported similar diurnal profiles for the Cornelia Fort Airpark in 1999.*

I would like to see (in the SI) the MPAN vs PAN and PPN vs PAN scatter plots. Are the slopes comparable to observations from other studies? Can these slopes be used as characteristic signatures of anthropogenic and biogenic influences?

Roberts used a range of MPAN vs PAN depending on time of day. Did the authors have to do that as well or were the slopes constant throughout SOAS?

*We added the MPAN vs PAN and PPN vs PAN scatter plots in Figure S5 of the Supplementary Information. While slopes from a single linear regression (SLR) may be similar to the mean ratio, in the multiple linear regression (MLR), the t-value, which is obtained by partial slope/standard error, or β-value (standardized partial regression coefficient) is used to compare the influence of each independent variable instead of slopes, because slope calculations are different in MLR and SLR. In this study we used the determination of coefficients to determine anthropogenic and biogenic sources based on the β-value and correlation coefficient. In general, MLR is more accurate than SLR to predict dependent variable: the coefficient of determination, R$^2$ in MLR is higher than in SLR.*

*We limited the analysis in this manuscript to daytime data only, and similar to Roberts et al, 2002, we did find different slopes of the regression of MPAN on PAN on different days. The overall slopes of MPAN/PAN are in the same range as previous results, although SOAS has a greater slope overall than Dickson. The observed levels of PPN and MPAN were lower than seen in sites in the 1990s in the southeast such as, Nashville, Dickson, and Youth In., and limited number of data above detection limit made it challenging to compare the behavior at specific times. Since we used measurements from the entire campaign, the data has wider variance. The statistics should be able to assess each partial slope based on calculated probability (p-value). The p-values in Table S2 support the calculated values as signatures of anthropogenic and*

*biogenic influences. Hence, we think the results from MLR statistic are appropriate to discuss in this study.*

I would like to see (possibly appended to figure 1) a diurnal plot of the MLR calculated PAN next to the observed PAN. The MLR model explains 60% of the variance. On average, when is the agreement good and bad. I am skeptical how robust the anthro vs biogenic attribution is for SOAS given how PAN, PPN and MPAN appear all well correlated in time (figure 1).

*We added the plot for measured PAN versus predicted PAN using MLR in Figure S6. The diurnal plot of predicted and measured PAN was added in Figure S7. On average, predicted PAN correlated well with measured PAN, especially in the afternoon. However, the standard deviation of both measured and predicted data are similar (that is why p-value from ANOVA test in Table S1 showed less than significant level). This gives us confidence that the coefficient values for PPN and MPAN in MLR are representative (or robust) values for overall campaign.*

Section on attribution of BHC derived PAN to isoprene. For the ODR approach, your only input was from chamber oxidation of isoprene. Were you able to test MPAN production rates from MBO oxidation? Monoterpenes oxidation? I am concerned that your answer came out to be isoprene only because your input to the ODR was isoprene.

*As the reviewer points out, the ODR approach in this study is limited, since we used isoprene as a proxy for biogenics. This is not completely unreasonable. Isoprene accounted for the majority of OH reactivity at SOAS (Kaiser et al., 2016), MBO has not been shown to produce MACR, at least from OH and $O_3$ chemistry (e.g. Alvarado et al., 1999; Spaulding et al., 2003) and some monoterpenes can form PAN as a secondary product, but not MACR. The ODR method compared a smog-chamber derived ratio of MPAN/PAN from isoprene to a modeled ratio and used that ratio to get an approximation of isoprene attribution, which is a lower limit of total BVOC attribution.*

I question the relevance/validity of discussion section 4.1. Authors used MPAN vs PAN to establish that 50-70% of SOAS PAN is biogenic. You determined that biogenic PAN is mostly from isoprene. So why is it surprising that MPAN and isoprene hydroxy nitrates are well correlated?

*We expected that primary products from isoprene oxidation (MACR and IN) would be more tightly correlated than a primary and a secondary (MPAN and IN). So maybe "surprising" is too strong of a word, but it is noteworthy that. Also we added a figure of the ratio of IN/MPAN as a function of NOx that shows that the variability of the relationship is different under different NOx conditions. In the range of NOx that Thornton et al. (2002) found for high P(HOx) the ratio spans a large range. At higher NOx values the ratio is relatively constant.*

The statement that "...MPAN shows significant contribution to OA growth...", based entirely on correlation, is unconvincing. The contribution to OA from MPAN can be estimated knowing the volatility (or C*) and its abundance in the gas phase or its SOA yield along with its lifetime. More data (calculations, model runs, etc.), not citations to other papers that merely suggest the possibility, is needed.

*The word "significant" in this phrase simply refers to the statistical test of the correlation and does not mean to imply a specific contribution to aerosol growth. We agree that the data are too limited to draw strong conclusions, but they do support the hypothesis that there is a contribution to aerosol growth without inclusion of nitrogen. We have re-worded this statement to remove the word significant.*

Need citations for sentence on lines 16-18, page 10.

*Three representative citations were added: Paulot et al. (2009), Mao et al. (2013), and Liu et al. (2013).*

**Suggestions from and responses to Anonymous Referee #2**

General comments:

Section 1: The first paragraph of the introduction should be placed in the experimental section as it is related to the measurements site and context. It would be useful to give typical mixing ratios (or ranges) of PANs expected in urban/rural/forested environments with information on seasonal variations and global trends over the last decades. The different sinks of PANs should be described. The introduction should better underline the importance of understanding PANs chemistry (NOx removal/ transport, aerosol aging).

*This first paragraph was moved to Section 2, as suggested. The introduction was enhanced to give a more detailed evaluation of the importance of PAN in the global atmosphere and to outline the loss mechanisms for PAN compounds as suggested by the reviewer.*

Section 2: The experimental section should contain a brief description of the meteorological conditions at the site during the campaign (temperature, humidity ...). Were those conditions expected at this lo-cation or were there different from classically encountered conditions? A citation could be placed if other papers describe the campaign in more details. I don't understand the point of discussing in such details the sum of PANs measurements by thermal dissociation as those data are used marginally in the paper. The authors should precise what they mean by NOy and NOy measurements. This last comment apply to the whole manuscript.

*The meteorological conditions of the ground site have been described in detail in Carlton et al, 2018 and Hidy et al., 2014. We have indicated this in the revised version. The point of discussing the comparison of measurements is for due diligence of somewhat co-located measurements at the site, although there was no formal inter-comparison between the measurements during the campaign. This gives us limits on the confidence we have for the speciated measurements. We agree that having this section in the main paper can be distracting and we have moved it to the supplemental materials. NOy is defined as gas-phase oxidized nitrogen now in the experimental section where it is introduced.*

Section 3.1: The time series of PANs (figure 1) should be accompanied by temperature, NOx and O3 as correlations are strongly expected and this would help to see anthropogenic influence. The profiles of PANs could be discussed further. To what is due the morning peak, is it advection? Mixing with residual layer? It would be useful to add a profile for global radiation to figure 2. Since the sum of PANs have been measured, it would be interesting to know what fractions of total PANs represent PAN, PPN and MPAN. This would justify the big paragraph of total PANs measurements comparison in the experimental section.

*We have added timeseries for NOx, O₃, and Temperature to Figure 1. The diurnal morning increase observed for PANs is also observed in ozone, NOx, and isoprene is a result of morning breakup of a nocturnal inversion that was commonly observed at the site. It is very similar to variations observed at other ground sites (Nashville, Dickson, BEARPEX, PROPHET). We have also included in Table 1 additional information regarding the relative concentrations of various parameters, as suggested by the reviewer, for the SOAS site as well as for the other historical sites where PANs were measured. This was made more clear in the text.*

Section 3.2: Figure 3 is very interesting and show well that the 2013 measurements might correspond to a shift to a NOx limited regime, however, the authors should discuss further the possibility that the observations on the 2013 can be explained by a lower PHOx (due to lower photolysis rates or OH precursors) as mentioned very/too briefly and explained in Thornton et al. 2002. The authors should also discuss the possibility that observations correspond to older air masses or higher NOx/PANs loss rates compared to the other measurements which would explain the lower PAN and NOx data.

*This is a good point. As seen in Figure 1, and explained in Hidy et al., ozone, NOx, and photolysis rates were all lower at this ground site than in previous years, which likely results in lower overall oxidation rates due to lower radical production rate (PHOx). This paragraph has been revised.*

Section 3.3: The authors should describe briefly the hypothesis that are necessary to apply the MLR (and not only cite references) and the validity limits. Especially, this method imply that all PANs are only lost by thermal decomposition, however, the authors state in Section 4 that the reaction MPAN + OH represent a non-negligible sink of MPAN. Doesn't it invalidate/limit the MLR analysis? LaFranchi et al. 2009, using a steady state method for PAN sources attribution, show that the results (relative parts of BHC and AHC) are strongly dependent on temperature, because this factor affects the emissions of isoprene (Worton et al. 2013). This could maybe explain the differences between the 2013 and 1999 results (if temperature was much higher in 2013). The authors should comment on this. Why not apply all 3 methods to all the historical measurements? This would allow to make the conclusions more robust if for all rural measurements in the 90s, the BHC role as PAN precursors was higher than for urban measurements, and if all rural measurements in the 90s were all more oriented toward AHC than the 2013 results. This would maybe allow to conclude on the decrease of AHC role in PANs production.

*The MLR is a purely observational model that necessarily incorporates all loss processes for PANs active at the time and place of the measurements. Because OH reactions with MPAN is more important than with PAN or PPN the biogenic (MPAN) contribution might be conservative in our analysis. As pointed out, it appears that P(HOx) was low at SOAS 2013, so this factor may be smaller than at other sites. Additional modeling is beyond the intention of this work, and application of other models at Dickson is hampered by the more limited dataset available for Dickson and other sites.*

Section 4: As stated by the authors, the relationship between IN and MPAN should be depending on the NO/NO2 ratio. This fact could be well visualized by a plot of IN vs MPAN color coded with the NO/NO2 ratio. The authors cite Worton et al. 2013 but do not mention that in this reference, the authors state that MPAN uptake on aerosols results in the formation in organo-sulfates in the aerosol pahse, which is a likely explanation for the weak dependence of pONs on MPAN.

*As recommended by the reviewer, a graph of the ration IN/MPAN vs NOx has been included as Figure 6 in the manuscript. The figure shows two modes. The IN/MPAN ratio varies over a large range when NOx <1ppb, corresponding to the range of high P(HOx) described by Thornthon et al., (2002). At higher NOx levels, the ratio remains fairly constant around 2.5-3. At lower NOx levels, NOx is more efficiently tied up in IN than in MPAN) and the MPAN+OH loss rate at low NOx? We have edited the paragraph to explain this. Reference to the organosulfate mechanism has been added to this section. Thanks.*

Section 3 and 4: Isoprene has a central role in all results and discussions but nothing is said about its mixing ratios which were measured in 2013 and 1999 together with the PANs.

*In the supplemental materials we have added figures illustrating isoprene levels at SOAS and Dickson in Figure S4b. Emissions at both sites were significant, but with lower photolysis rates and the lower levels of radical precursors (mainly O₃) the lifetime of isoprene was likely greater at SOAS leading to higher average daytime values.*

*Most of the "Specific Comments" made by reviewer #2 listed below were incorporated in the revised manuscript exactly as suggested.*

Specific comments page 1 line 31 : sensitive could be replace by "controled by NOx"

*Replaced.*

page 2 line 1 & 2 : the last sentence is not really necessary

*Deleted.*

page 2 line 9 : also give official IUPAC name

*Added.*

page 2 line 11 : The role and importance of PANs could be placed in a global context instead of being eeduced to the Eastern U.S.

*Added several sentences to expand the context.*

page 2 line 13 : phytotoxic

*This sentence was revised to read "they are significant health hazards for both humans and plants."*

page 2 line 14 : how abundant in term of fraction of PANs ?

*Added the approximate range of PAN/NOy values that have been observed.*

page 2 line 25 & 26 & 27 : this sentence is confusing, maybe removing the because would help. aerosol radiative forcing could be replaced by secondary organic aerosol formation.

*Revised as suggested.*

page 3 line 6 : "was" should be replaced by "is" ; which type of vegetation does "forested" correspond to ?

*Clarified.  A more detailed description of the forest type is given with reference.*

page 3 line 10 : what about BHC sources at the Dickson site

*Information about mean isoprene values for Dickson have been included in Table 1.*

page 3 line 15 : what about air mass origins during the measurements ?

*Air mass origins have been discussed in Hidy et al. (2014) and Carleton et al. (2018).*

page 4 line 3 : what is NOy, is it total NOy by catalytic conversion ?

*NOy was defined explicity as described in Hidy et al., 2014..*

page 4 line 15 : what does WMU stands for ?

page 4 line 18 : Where are those WMU measurement described ? With which type of instrument ?

*These two comments refer to the comparison between measurements that was moved to the supplemental materials.  WMU stands for Western Michigan University, where the primary author did the work.*

page 4 line 27 & 28 : the correlation allows for the investigation of PANs behavior but those values are never used in this paper, why compare the 3 instruments then ?

*As described above, the point of discussing the comparison of measurements is for due diligence of somewhat co-located measurements at the site, although there was no formal inter-comparison between the measurements during the campaign.*

page 5 line 7 : add "(see Sect.3.2)" after "the last 20 years"

*Added.*

page 5 line 9 : what about the 2 others peaks near 1 ppbv

*The other peaks could not be attributed to extremely local biomass burning.*

page 5 line 11 : what is the ratio between PAN and sum of PNs ? page 5 line 14 : define NOy

*Details for all sites can now be seen in Table 1. We also included a line in the text that reads "PAN was consistently the most abundant peroxyacyl nitrate compound, the mean daytime levels accounting for approximately 90% of total PANs."*

page 5 line 20 : replace "surface air" by "air masses" or "sampled air" ; they seems to be a net differ-ence between air masses from the south and air masses from the north. Would the MLR for PAN pre-cursors identification reveal a difference as well between south and north.

*Surface air was replaced.  We also included a polar plot of isoprene concentrations in the supplemental materials that shows the biogenic precursors are not as directionally dependent.*

page 5 line 24 & 25 : it would be nice to be able to see NOx and O3 somewhere. And what about differences in terms of VOCs mixing ratios between North and South, some BHC and AHC were measured during SOAS 2013. They could be described somewhere in the paper.

*These have been added to Figure 1..*

page 5 line 29 : to solve that a different scale could be used on figure 2 for the North and the South data

*This sentence was revied to rea" The diurnal cycle for PPN was less pronounced because of the low concentrations observed over the campaign".*

page 5 line 31 : maybe an equation would be helpful for the decomposition lifetime calculation.

*An equation has been added to the Introduction section to describe PAN loss dependence on NOx.*

page 6 line 11 : replace "higher PAN concentration with higher NOx" with "higher PAN and NOx concentrations"

*Replaced.*

page 6 line 12 : what does "revisited" means ?

*The phrase was modified to read "The only site sampled in more than one year was ROSE…".*

page 6 line 13 : "hence, the PAN concentrations can vary depending on place and year" could be replaced by "Overall, the PAN concentrations were strongly variable between sites and years" ; which type of curve fit ?

*The sentence was revised. A log-normal curve fit function is now indicated.*

page 6 line 17 : "and the peak was at around" should be replaced with "with a maximum around"

*Replaced.*

page 6 line 30 : yes covariance has been observed but that is because they are both produced by pho-to-oxidation of VOCs in the presence of NOx, this is the reason why PAN vs NOx looks similar to O3 vs NOx : their production pathways are the same. \

page 7 line 2 & 3 : the sentence "and most PAN concentration at rural sites were dependent on NOx concentrations" is confusing and does not bring any information. It seems to say that only in rural areas are PAN concentrations correlated with NOx concentrations which is not the case.

*We thank the reviewer for pointing out the oversimplification of the last 2 comments. We have revised the text to read "The covariance between PAN and $O_3$ (Bottenheim et al., 1994) due to their common photochemical pathway in the atmosphere suggests that the steep increase of PAN concentrations with $NO_x$ at low $NO_x$ in Figure 3a could result from $NO_x$-limited chemistry. Most rural sites showed PAN levels more sensitive to $NO_x$ concentrations."*

page 7 line 7 : references ?

*References to decreases in NOx concentrations and emissions have been added.*

page 7 line 11 : "sources" should be replaced with "precursors"

*Replaced.*

page 7 line 18 : it should be mentioned somewhere that the A factor correspond to background PAN

*A parenthetical phrase was added.*

page 8 line 16 : replace "Also, in Dickson 1999 . . . higher" with : "while NOx levels were seven times higher".

*Replaced.*

page 8 line 19 : "(mostly isoprene)" could be added behind "Biogenic influence" C6

*Added.*

page 9 line 8 : describe the 0D model method in more details. What are the hypothesis?

*The 0D model was based on the Master Chemical Mechanism v3.3. Explicit isoprene chemistry together with the inorganic mechanisms was used included chemical mechanisms for some monoterpenes (a-pinene, b-pinene,and limonene) was used to evaluate ambient field data. PANs were the main focus, so the model was constrained with a subset of parameters ($NO_2$, NO, OH, $HO_2$, CO, $H_2O$, ozone, acetaldehyde, and acetone). In addition to thermal and OH loss processes, the model included PAN loss by deposition.*

page 10 line 25,26,27 : the fact that IN is high during the day does not mean that its production is high during the day, it could be produced by NO3 oxidation of isoprene and have a long lifetime enough to be observed during the day, which is why talking about daytime in line 26 is not very accurate. Moreover, saying that IN is the dominant sink is as well not accurate. IN is a sink of NOx if IN removal leads to a net loss of NOx, but what happens if IN releases NOx due to oxidation or due to uptake to the aerosol phase and subsequent release of NOx. How does the general context of those measurements compare to Romer et al. ?

*We agree this section was overstated, and again thank the reviewer for calling it out. We have toned down the text by removing the last sentence. We discuss the NOx dependence in more detail, as described earlier, with a new figure. Romer et al. do find that the production and loss of organic nitrates strongly affect NOx at SOAS and that daytime production is significant.*

page 11 line 4 : Worton et al. 2013 suggest that uptake of organics following MPAN + OH reaction occurs through the formation and subsequent uptake of methacrylic acid epoxide (MAE).

*A reference to Worton et al. 2013 was included in this sentence.*

page 12 line 5 : remove "data"

*Removed.*

page 12 line 6 : you showed than PAN production is limited by NOx availability,

*The sentence was revised as suggested.*

page 12 line 8 : what is seen is that lower NOx emissions seem to result in lower ambient PAN concentrations.

*The sentence was revised as suggested.*

page 12 line 11, 12, 13 : the first part of the sentence just repeats line 6 and 7, the second part of the sentence is confusing, where is MPAN production rate as a function of NOx discussed ?

*The sentence was removed to avoid redundancy.*

page 12 line 16 : same remark as for line 25, 26, 27 page 10, it is not clear that IN is a net sink, since the removal pathway that is discussed in this paper, aerosol uptake, does not seem to trap NOx in the aerosol phase.

*The last part of this sentence suggesting that IN is a net sink was removed.*

page 12 line 17 : is 66% an average of the three methods ? The sentence "twice as much as anthropogenic influence during the overall campaign" is redundant. If biogenic influence is 66%, then the rest is obviously anthropogenic influence and logically 33% which is ... twice less.

*Redundancy was removed.*

**Importance of Biogenic Volatile Organic Compounds to Peroxyacyl Nitrates (PANs) Production in the Southeastern U.S. during SOAS 2013**

**Shino Toma[1], Steve Bertman[1], Christopher Groff[2], Fulizi Xiong[2], Paul B. Shepson[2], Paul Romer[3], Kaitlin Duffey[3], Paul Wooldridge[3], Ronald Cohen[3], Karsten Baumann[4], Eric Edgerton[4], Abigail R. Koss[5,7*], Joost de Gouw[5], Allen Goldstein[6], Weiwei Hu[7,8], and Jose L. Jimenez[7,8]**

[1]{Department of Chemistry, Western Michigan University, Kalamazoo, MI, USA}
[2]{Departments of Chemistry, and Earth, Atmospheric, and Planetary Sciences, Purdue University, West Lafayette, IN, USA}
[3]{Department of Chemistry, University of California, Berkeley, CA, USA}
[4]{Atmospheric Research & Analysis, Inc., Cary, NC, USA}
[5]{NOAA ESRL Chemical Sciences Division, Boulder, CO, USA}
[6]{Department of Environmental Science, Policy and Management, University of California, Berkeley, CA, USA}
[7]{Cooperative Institute for Research in Environmental Sciences, University of Colorado, Boulder, Colorado, USA}
[8]{Department of Chemistry and Biochemistry, University of Colorado, Boulder, Colorado, USA}
*now at Department of Civil and Environmental Engineering, Massachusetts Institute of Technology, Cambridge, MA, USA

Correspondence to: S. Bertman (steven.bertman@wmich.edu)

KEY WORDS: isoprene, PAN, MPAN, SOAS, BVOC

**Abstract**

Gas-phase atmospheric concentrations of PAN, PPN, and MPAN were measured at the ground using GC-ECD during the SOAS 2013 campaign (1 June to 15 July 2013) in Centerville, Alabama in order to study biosphere-atmosphere interactions. Average levels of PAN, PPN and MPAN were 169, 5, and 9 pptv respectively, and the sum accounts for an average of 16% of $NO_y$ during the daytime (10 am to 4 pm local time). Higher concentrations were seen on average in air that came to the site from the urban $NO_x$ sources to the north. PAN levels were the lowest observed in ground measurements over the past two decades in the Southeastern U.S. A multiple regression analysis indicates that biogenic VOCs account for 66% of PAN formation during this study. Comparison of this value with a 0-D model simulation of peroxyacetyl radical production indicates that at least 50% of PAN formation is due to isoprene oxidation. MPAN has a statistical correlation with isoprene hydroxynitrates (IN). Organic aerosol mass increases with gas-phase MPAN and IN concentrations, but the mass of organic nitrates in particles is largely unrelated to MPAN.

**1. Introduction**

Peroxyacyl nitrates (carboxylic nitric anhydrides) (PANs, RC(O)OONO$_2$), products of the photooxidation of VOCs in the presence of nitrogen oxides (NO$_x$), play an important role in the chemistry of both gases and particles in the troposphere. Measurements around the world have shown that PANs can comprise 15-40% of total gas-phase oxidized nitrogen in rural and forested areas, including the Eastern United States depending on how aged the air sampled at the site (e.g. Trainer et al., 1993, Nouaime et al., 1998). The fraction depends on air mass history and conditions. Gas-phase nitric acid, the other major component is more easily lost than PAN so HNO$_3$ deposition rate is also a factor. 
[revised manuscript text omitted]

[Figure]

(a)

[Figure]

(b)

Figure S4. Hourly diurnal profiles of mean (a) PANs/$NO_y$ and (b) isoprene with one standard deviation
from the Dickson 1999 and SOAS 2013 campaigns.

**Methods and Results of MLR analysis for PANs**

$$[PAN] = A + B_1[MPAN] + B_2[PPN] \tag{1}$$

In a multiple linear regression (MLR) model as Equation (1), [PAN] is treated as a response variable and [MPAN] and [PPN] are used as independent predictor variables. $B_1$ and $B_2$ are partial regression coefficients on [MPAN] and [PPN]. The MLR statistical analysis conducted two steps of statistical testing. First, the $F$-test in ANOVA and $R^2$ investigated how well the model Eq. (1) fits the measurement data. However, $F$-test is impossible to directly find out which predictor variable is significantly useful. Therefore, in the next step, the significant utility of each partial regression coefficient was explored using the Student's *t*-test. The respective *t*-value was calculated from each partial regression coefficient divided by the standard error. When results of the *t*-test indicate presence of statistical significance for the partial regression coefficients, the magnitude of the standardized partial regression coefficient, $\beta_i$, allows us to compare the relative contribution of each independent predictor variable within the model.

As the notice to conduct MLR statistical analysis, high multicollinearity causes effects on the results of the analysis (e.g. Mendenhall et al., 2009). Although the assumption of the MLR statistical analysis on [PAN] takes a stance that each predictor variable is derived from different hydrocarbon precursor independently, the values of "tolerance" or "variance inflation factor (VIF)" were helpful to assess the impact of the multicollinearity. The tolerance is calculated as 1- $R^2_{MPAN-PPN}$, where $R^2_{MPAN-PPN}$

is the coefficient of determination between MPAN and PPN and VIF is 1/tolerance. Large VIF value indicates strong multicollinearity of predictor variables. According to Stevens (2012), if the value of VIF

is greater than 10, it indicates effective multicollinearity.

The statistical analysis was conducted using SPSS statistics software (versions 16, IBM). Results of *F*-test and $R^2$ on the MLR model for SOAS 2013 during the daytime are summarized in Table S1.

Similar PANs data collected from Dickson, TN during the SOS experiment in 1999 is used as a comparable reference. The small *p*-value (*P* in Table S1) of *F*-test indicated that the overall fit of the model Eq. (1) is statistically significant in both the SOAS 2013 and Dickson 1999, and at least one independent predictor variable was significantly useful.

Table S1. Summary of the *F*-test and $R^2$.

| Year | Number of data | *P* of *F*-test | *R* | $R^2$ |
|---|---|---|---|---|
| Dickson, TN in 1999 | 486 | <0.001 | 0.876 | 0.766 |
| SOAS 2013 | 498 | <0.001 | 0.775 | 0.601 |

A summary of coefficients of MPAN and PPN in both SOAS 2013 and Dickson 1999 is shown in

Table S2. Since all VIF values were less than 10, there was no impact of multicollinearity in the MLR

statistical analysis in both SOAS 2013 and Dickson 1999. The small *p*-value (*P* in Table S2) of the *t*-test of both MPAN and PPN in SOAS 2013 and Dickson 1999 indicates both predictor variables were useful to predict PAN. Therefore, respective partial regression coefficient values were available to estimate PAN

in SOAS 2013 and Dickson 1999.

Table S2. Summary of coefficients on each independent predictor variable in *t*- test.

| | Dickson, TN in 1999 | | SOAS 2013 | |
| | MPAN | PPN | MPAN | PPN |
|---|---|---|---|---|
| Partial regression coefficient | $B_1$ 5.098 | $B_2$ 5.762 | $B_1$ 7.596 | $B_2$ 6.910 |
| Std. error of coefficient | 0.305 | 0.178 | 0.469 | 0.725 |
| $P$ of *t*-test | <0.001 | <0.001 | <0.001 | <0.001 |
| VIF | 1.036 | 1.036 | 1.427 | 1.427 |
| $\beta_i$ | 0.374 | 0.725 | 0.549 | 0.323 |
| $r_i$ | 0.509 | 0.795 | 0.726 | 0.624 |
| Partial $R^2 = \beta_i \, r_i$ | 0.190 | 0.576 | 0.399 | 0.202 |

Std. error of coefficient means standard error of partial regression coefficient. *P* is calculated probability. $\beta_i$ is standardized partial regression coefficient. $r_i$ is zero-order correlation. All dataset was during the daytime, 10 am – 4 pm.

[Figure]

(a)                                                    (b)

Figure S5. Scatter plots for PPN vs. PAN and MPAN vs. PAN in (a) Dickson 1999 and (b) SOAS 2013
during the daytime, 10 am – 4pm. The below detection limit data were included at half of the detection
limit. The solid line is the fit for MPAN to PAN and the dash line is the fit for PPN to PAN. The slopes
with standard deviation were 0.037±0.003 ($R^2$=0.259) for MPAN to PAN and 0.100±0.003 ($R^2$=0.633)
in Dickson 1999, and 0.053±0.002 ($R^2$=0.530) for MPAN to PAN and 0.029±0.002 ($R^2$=0.390) in SOAS
2013.

[Figure]

(a)  (b)

==Figure 6. Measured PAN versus predicted PAN using MLR statistic (a) in Dickson 1999 and (b) in==
==SOAS 2013 during the daytime, 10 am – 4 pm.== The below detection limit data for PPN and MPAN in
SOAS 2013 were included at half of the detection limit in this experiment to avoid to lose the low
concentration information. (JMP version 12.1.0, SAS Institute Inc.)

[Figure]

==Figure S7. Diurnal plot of measured PAN in SOAS 2013 and predicted PAN using MLR statistic.==
(Note: this measured PAN was not filtered by wind direction like Figure 2.) Predicted PAN was
calculated based on measured PPN and MPAN during the daytime (10 am – 4pm).

---

## Author Response (AR2)

**Responses to the editor**

*Thank you so much for your excellent contributions to improve our paper. Revisions in the manuscript (as described below) in response to your questions are highlighted with light blue color in the Word file for easy reference (we kept the changes based on reviewers' comments in yellow).*

**Co-Editor Decision: Publish subject to minor revisions (review by editor)** (26 Nov 2018) by Jennifer G. Murphy
Comments to the Author:
The authors have generally done a good job in addressing the concerns of the reviewers. I have a few remaining suggestions prior to publication in ACP.

Section 3.3.1 – I am still confused about how to interpret the MLR statistics.
Are the 'standardized relative importances' presented in Figure 4 meant to be interpreted as 66% of PAN comes from biogenic precursors, as the text seems to suggest? In other words, in 2013, twice as much PAN comes from biogenic precursors compared to anthropogenic precursors.
*This interpretation is based on the coefficient of determination, $R^2$, between predicted PAN and measured PAN. We partialized the $R^2$ for each independent variable as Eq. (3) and (4) (and standardized by total $R^2$ in Figure 4). We revised this section.*

Or is it that twice as much of the variability in the amount of PAN is explained by variability in biogenics compared to variability in anthropogenics?
*Since $R^2 = \Sigma(\hat{y}_i - \bar{y})^2 / \Sigma(y_i - \bar{y})^2 = 1 - \Sigma(y_i - \hat{y}_i)^2 / \Sigma(y_i - \bar{y})^2$, where $y_i$ is measured PAN, $\hat{y}_i$ is predicted PAN, and $\bar{y}$ is average of measured PAN, and $\hat{y}_i$ depends on the variability of dependent variables, $R^2$ is technically related to variability of dependent variable.*

Page 2, L16 – NO is not reduced in this reaction, it is oxidized
*Modified to "reduction **by** NO".*

Page 2, L17-18 – Would make more sense in the context of the paragraph to say that 'Peroxy radicals compete with NO…'
*Replaced as suggested.*

Page 2, L18-19 The content in the boxes should be reformatted as reactions (R1,-1 and separately R2) and the equation should be listed separately as Eq 1
*Reformatted as suggested.*

Page 6, L21-23 – This sentence is worded strangely. Why would RO2+NO reactions becoming faster make ozone production decrease? Isn't it that NOx radical termination reactions become

the dominant HOx sink?

*We reworded our description to better reflect your point that radical termination steps that remove NOx from the system are heightened under high NO conditions, namely formation of nitric acid and organic nitrates.*

Page 7, L6-7 "Most rural sites…" Do the authors mean that PAN was more sensitive to NOx at sites with low NOx, which tended to be rural? The sentence is not clear as written.

*Yes, it does. Clarified as suggested to read "PAN at sites with lower $NO_x$ levels seem to be more sensitive to $NO_x$ concentrations, as is seen for most of the rural sites in this region."*